# Decreased Podocyte Vesicle Transcytosis and Albuminuria in *APC* C-Terminal Deficiency Mice with Puromycin-Induced Nephrotic Syndrome

**DOI:** 10.3390/ijms222413412

**Published:** 2021-12-14

**Authors:** Saaya Hatakeyama, Akihiro Tojo, Hiroshi Satonaka, Nami O. Yamada, Takao Senda, Toshihiko Ishimitsu

**Affiliations:** 1Department of Nephrology & Hypertension, Dokkyo Medical University, Tochigi 321-0293, Japan; saaya.hatakeyama@gmail.com (S.H.); satonaka@dokkyomed.ac.jp (H.S.); isimitu@dokkyomed.ac.jp (T.I.); 2Department of Anatomy, Gifu University, Gifu 501-1193, Japan; nyamada@gifu-u.ac.jp (N.O.Y.); tsenda@gifu-u.ac.jp (T.S.)

**Keywords:** podocyte, vesicle transport, microtubule, nephrotic syndrome, adenomatous polyposis coli, glomerular filtration barrier

## Abstract

In minimal change nephrotic syndrome, podocyte vesicle transport is enhanced. Adenomatous polyposis coli (APC) anchors microtubules to cell membranes and plays an important role in vesicle transport. To clarify the role of APC in vesicle transport in podocytes, nephrotic syndrome was induced by puromycin amino nucleoside (PAN) injection in mice expressing APC1638T lacking the C-terminal of microtubule-binding site (APC1638T mouse); this was examined in renal tissue changes. The kidney size and glomerular area of APC1638T mice were reduced (*p* = 0.014); however, the number of podocytes was same between wild-type (WT) mice and APC1638T mice. The ultrastructure of podocyte foot process was normal by electron microscopy. When nephrotic syndrome was induced, the kidneys of WT+PAN mice became swollen with many hyaline casts, whereas these changes were inhibited in the kidneys of APC1638T+PAN mice. Electron microscopy showed foot process effacement in both groups; however, APC1638T+PAN mice had fewer vesicles in the basal area of podocytes than WT+PAN mice. Cytoplasmic dynein-1, a motor protein for vesicle transport, and α-tubulin were significantly reduced in APC1638T+PAN mice associated with suppressed urinary albumin excretion compared to WT+PAN mice. In conclusion, APC1638T mice showed reduced albuminuria associated with suppressed podocyte vesicle transport when minimal change nephrotic syndrome was induced.

## 1. Introduction

The first electron microscopy observation of glomerulus in 1955 prompted several theories concerning glomerular filtration involving detection of the slit membrane as a glomerular filtration barrier [1] and transepithelial transport via podocyte vesicles [2]. Unfortunately, the concept of transepithelial transport via podocyte vesicles has long been neglected. The glomerular basement membrane has been recognized as an essential barrier [3]. The observation of a zipper-like structure in the slit membrane [4] followed by the detection of nephrin as a slit membrane molecule [5] confirmed the concept that albumin leaks through the slit membrane pores in nephrotic syndrome [6,7]. Later, progression of proteinuria in laminin beta 2 mutant mice again reminded us of the importance of GBM as a glomerular filtration barrier [8,9]. The mystery of the glomerular filtration barrier (GFB) is why it does not clog [10] when filtering 3 g of albumin daily in the primary urine, even under normal conditions [11]. Gel hypotheses and electrokinetic models have been proposed to solve the mystery of GFB clogging [10,12], but not enough attention has been paid to the mechanism of transcytosis in podocytes.

We reestablished the concept of albumin filtration through the podocyte cell body in minimal change nephrotic syndrome (MCNS) [13] and demonstrated the selective albumin transport in podocytes via FcRn receptors [14]. Several albumin transport mechanisms, including clathrin-mediated endocytosis [15], megalin-mediated endocytosis [16], caveolae-mediated endocytosis [17] and macropinocytosis [18], have been described [19]. Electron microscopic observation of podocyte of MCNS patients showed numerous vesicles [19,20,21], and glomeruli under conditions of MCNS showed increased motor proteins, including cytoplasmic dynein 1, myosin IXa (Myo9a) and myosin VIIb (Myo7b) [22]. Cytoplasmic dynein transports cargo along the microtubules toward the minus end, attached under the adhesion junction [23,24]; podocyte vesicle transport may thus explain selective albuminuria in MCNS [19,22].

Adenomatous polyposis coli (APC) is a tumor suppressor protein in colon cancer [25] and an anchor protein that binds to β-catenin with postsynaptic density postsynaptic density (PSD)-93/95 in the adhesion complex of the synaptic membrane [26,27]. APC also binds to both the actin and microtubule plus-ends [28], thereby positioning a microtubule-based transport pathway to carry receptor proteins to postsynaptic regions [27]. APC1638T mice express a mutant APC protein lacking the C-terminal side after 1639 amino acids, preventing binding with PSD-95, discs large (DLG), microtubules and end-binding protein 1(EB1) [29,30,31,32]. Glomerular podocytes contain similarities to neurons and express PSD-93 [33], so in the present study, we investigated the morphological changes in podocyte microtubule and vesicle transport in APC1638T mice with PAN-induced MCNS and discussed the importance of podocyte vesicle transport as a mechanism of selective albuminuria.

## 2. Results

### 2.1. Kidney Size, Glomerular Volume, and Podocyte Number and Structure in APC Mutant Mice

The kidney of APC1638T mice was examined by PAS staining, and the size of the kidney and glomerulus seemed to be smaller than in wild-type mice (Figure 1). The size of the glomeruli and number of podocytes in each glomerulus were calculated using semi-thin sections of EPON-embedded tissue stained with 0.05% Toluidine Blue solution. The glomerular cross-sectional area of APC1638T mice was smaller than that of wild-type mice (*p* = 0.014); however, the number of podocytes was not significantly different between wild-type (WT) mice and APC1638T mice (Figure 2). The ultrastructure of the podocyte did not differ markedly between WT and APC1638T mice, and the foot process was preserved in APC1638T mice (Figure 3).

### 2.2. Kidneys of APC Mutant Mice with PAN-Induced Nephrotic Syndrome

The kidney in WT+PAN mice was larger than in APC1638T+PAN mice, and many hyaline casts were observed in WT+PAN mice, a finding that was suppressed in APC1638T+PAN mice (Figure 4). Electron microscopic observation showed that podocyte vesicles were located through the cytoplasm of podocytes in WT+PAN mice, even though there were more around the Golgi apparatus, whereas significantly fewer endocytosis vesicles were distributed around the basement membrane area; there were also significantly fewer exocytosis vesicles around the apical membrane area, and enlarged tubular structures were observed around the Golgi apparatus in podocytes in APC1638T+PAN mice compared to WT+PAN mice (Figure 5), suggesting decreased podocyte vesicle transcytosis.

### 2.3. Cytoplasmic Dynein and α–Tubulin Expression in the Podocyte and Albuminuria in APC Mutant Mice with Nephrotic Syndrome

Immunostaining revealed that cytoplasmic dynein-1 and α-tubulin were expressed mainly in the podocytes in glomeruli, and the products of the area and density of immunoreactivity of podocytes were significantly higher in WT+PAN mice than in APC1638T+PAN mice (Figure 6).

Real-time polymerase chain reaction (PCR) of whole-kidney homogenate demonstrated that α-tubulin mRNA was significantly increased in WT+PAN mice compared to WT mice, whereas it did not increase in the kidneys of APC1638T+PAN mice compared to APC1638T mice (Figure 7). Cytoplasmic dynein-1 did not show a significant change in the whole kidney among four groups, probably because it was expressed more strongly in the tubules than in podocytes (Figure 7). Urinary protein excretion did not differ between WT and APC1638T in the normal condition, whereas increased urinary albumin excretion in WT+PAN mice was significantly suppressed in APC1638T+PAN mice (Figure 8).

### 2.4. Immunofluorescence of Nephrin and Podocin Expression

To exclude the possibilities of albumin filtration through slit membrane, we performed immunofluorescence of nephrin and podocin expression in the glomeruli. There was no difference in the expression of nephrin and podocin in the glomeruli of WT and APC1638T mice; however, both mice showed reduction of nephrin and podocin in the glomeruli (Figure 9) associated with foot process effacement (Figure 5). As markers of apoptosis, caspase 3, a component of Fas death-inducing signaling complex, and Fas-Ligand were evaluated by immunofluorescence. Except for slight staining for caspase-3 in a few podocytes of WT+PAN mice, there was no significant staining for apoptosis markers in these mice.

## 3. Discussion

In the present study, we revealed that microtubule plus end anchor protein APC has an important role in the podocyte vesicle transport in the PAN-induced minimal change nephrotic syndrome, and APC1638T gene mutation reduced podocyte α-tubulin expression and reduced basal endocytosis vesicles, resulting in reduction of albuminuria in the PAN-induced nephrotic syndrome.

### 3.1. Renal Morphological Change in APC Mutant Mice under Control Conditions

The kidney size and glomerular cross section area were lower in APC1638T mice than WT mice, but the number of podocytes in the glomerulus was not significantly different (Figure 1 and Figure 2). As podocytes produce type IV collagen of the glomerulus via VEGF production and autocrine stimulation of TGF-β [34], the collagen produced in the Golgi apparatus was unable to be transported to the basement membrane where microtubule plus end could not bind with APC in the APC1638T mice, resulting in a reduced glomerular size. Indeed, electron microscopic observation revealed reduced basal vesicles in the podocytes of APC1638T mice (Figure 3). Podocytes are similar to neuronal cells and express PSD-93/95 [33]. APC forms a molecular complex with the NMDAR subunit NRA2 in the vesicles of neuronal cells and binds to microtubules and the kinesin kif3b, which transport the vesicle to the synaptic membrane. The kif3b mutation suppresses NMDAR containing vesicle trafficking to the spine surface membrane, resulting in a reduction in synaptic plasticity and development of schizophrenia-like symptoms [35]. APC also forms a molecular complex with nAChR in vesicles and plays a role in the transport of nAChR to the synaptic membrane [27]. The brain weight was shown to be heavier in APC knockout mice [35]. In contrast, APC1638T mice had a preserved β-catenin binding site in the center of the APC molecule, so cell proliferation via Wnt signaling can be blocked and thereby prevent cancer in APC1638T mice [30,31]. The body weight of APC1638T mice was significantly lower than that of WT mice at all ages [36], correlating with the smaller kidney and glomeruli sizes in APC1638T mice than in WT mice.

### 3.2. Morphological Characteristics of Podocyte of APC Mutant Mice with PAN-Induced Nephrotic Syndrome

APC binds to the plus end of the microtubules and F-actin located in the basal plasma membrane and is involved in cytoskeleton regulation in highly polarized epithelial cells [28,37]. In contrast, Ninein binds to the minus end of the microtubules near the apical membrane [38]. Under normal conditions, there were no obvious changes in the podocytes of APC1638T mice. When nephrotic syndrome was induced in WT mice, endocytosis vesicles in the basal region of podocytes and transcytosis vesicles in the podocytes increased (Figure 5), which is consistent with our previous report of increased endocytosis vesicles in the podocytes in association with motor proteins, including cytoplasmic dynein-1, myosin 7 and myosin 9 [22]. Interestingly, endocytosis vesicles were reduced from basal membrane, and tubular structures were enlarged around the Golgi apparatus in APC1638T mice with PAN nephrotic syndrome in the present study (Figure 5). PAN is a transfer RNA mimetic that is incorporated into nascent polypeptides and inhibits translation and protein synthesis [39], resulting in the accumulation of incomplete polypeptides in the enlarged tubular structure around the Golgi apparatus, especially in APC1638T mice. The nerve growth factor (NGF)-induced transport of different cargos requires local synthesis of different dynein cofactors; the microtubule plus-end tracking protein APC binds with the dynein cofactor Lis1 to transport large vesicles, whereas Lis1 and p150(Glued) transport smaller signaling endosomes via axonal transport [40]. Similarly, in podocytes, the inability to bind the positive end of the microtubule to dynein was suggested to potentially suppress vesicle transport and thereby reduce the number of vesicles in the basal area of podocytes in the present study (Figure 10). Moreover, the greater reduction in α-tubulin and cytoplasmic dynein-1 in podocytes of APC mutant mice with nephrotic syndrome than in WT mice with nephrotic syndrome (Figure 6) may result in a decrease in vesicle transport and albuminuria in APC C-terminal mutant mice (Figure 10).

### 3.3. APC as a Therapeutic Target of Selective Microalbuminuria

The loss of negative charge and reduction in nephrin in the slit membrane with foot process effacement is a classical concept of selective proteinuria [41]; however, this concept cannot explain why albumin is selectively excreted without an increase in low-molecular-weight protein in MCNS. As shown in Figure 9, the expression of nephrin and podocine did not change between WT and APC1638T mice in the normal condition, but they were reduced in both mice after PAN-induced nephrotic syndrome induction, merely reflecting the reduction in the number of slit membranes by foot process effacement that was observed to the same extent in both mice (Figure 5). These results indicate that slit membrane cannot explain the reduction of selective albuminuria in the APC1638T mice with PAN-induced nephropathy. Moreover, apoptosis markers including caspase-3 and Fas-Ligand did not stain significantly in the podocytes of these mice (Figure 9), indicating that podocyte detachment may not have a major role in the PAN-induced nephrotic syndrome model. The recent concept of podocyte vesicle transcytosis via albumin receptors, including FcRn [14], megalin [16] and caveolin [17], or via macropinocytosis, [18] requires a vesicle transport system and motor proteins [19]. We have previously observed that these vesicles in the podocytes contained albumin by immunoelectron microscopy [13,14]. Blocking of albumin receptor via anti-FcRn antibody reduced albuminuria by about 50% in vivo [14]. Furthermore, blocking of megalin-mediated endocytosis by gentamycin reduced podocyte albumin endocytosis according to intravital imaging, but albuminuria was not evaluated [16]. Blocking caveolin-mediated albumin endocytosis via nystatin [17], blocking of fluid-phase endocytosis via the aquaporin inhibitor pCMB, and cytochalasin D inhibiting actin polymerization [18] reduced albumin endocytosis in culture podocytes, but no data concerning in vivo albuminuria were available.

In the present study, we showed that APC C-terminal-depleted mice had reduced albuminuria in vivo. Under conditions of PAN-induced nephrotic syndrome, α-tubulin and cytoplasmic dynein-1 levels were increased in podocytes in WT mice with enhanced albuminuria, which is consistent with the findings of our previous study [22]. In contrast, endocytosis vesicles were unable to bind to the microtubule plus end or cytoplasmic dynein cofactor, as the APC C-terminal was lacking [40], resulting in the reduction of podocyte vesicle transport containing albumin (Figure 10). The regulation of binding of APC with microtubules and dynein-1 in the podocyte can be a molecular target of treatment of selective albuminuria in MCNS. Further studies will be required to clarify the specific role of APC in podocyte vesicle transport in nephrotic syndrome.

## 4. Materials and Methods

### 4.1. Animal Experiments

A total of eight 1-week-old male mice expressing APC1638T lacking the C-terminal with a microtubule-binding site (APC1638T mice) and their background WT C57BL/6JJmsSlc mice (Japan SLC, Hamamatsu, Japan) were harvested in a laboratory of Gifu University [42]. Four mice of each group had *ad libitum* access to tap water and standard mouse chow. MCNS was induced by administering puromycin aminonucleoside (PAN, Sigma Chemical Co., St. Louis, MO, USA) at 500 mg/kg body weight (BW) to WT and APC1638T mice. After 7 to 14 days, urinary protein levels were checked by a dipstick test to confirm nephrotic syndrome; 24-h urine samples were collected using metabolic cages, and urinary protein was analyzed by sodium dodecyl sulfate-polyacrylamide gel electrophoresis (SDS-PAGE, Invitrogen Waltham, MS, USA) with densitometry using ImageJ 1.53a software (NIH, Bethesda, MD, USA). After anesthesia, the kidneys were removed, and half of the kidney was immersion-fixed with 10% formalin for PAS staining and immunohistochemistry, while the other half was fixed with 2.5% glutaraldehyde for electron microscopy (JEM-1011, JEOL, Tokyo, Japan).

### 4.2. Periodic Acid Schiff (PAS) Staining, Immunohistochemistry and Image Analyses

Paraffin-embedded tissues were cut to 2-μm thickness and subjected to PAS staining. Immunohistochemistry was performed as previously described [33,43]. The number of podocyte nuclei were counted in the glomeruli cut near the center. The specialist of renal morphology (AT) counted the nuclei of podocyte by Toluidine blue staining of EPON-embedded renal sections. Podocytes were easily identified by their location outside of capillary wall, which was also confirmed by immunostaining for podocalyxin, as we have shown previously [44]. Sections (2-µm thick) were incubated with polyclonal antibodies against cytoplasmic dynein-1 or α-tubulin (Abcam, Tokyo, Japan) at 1:200 dilution followed by incubation with HRP-conjugated anti-rabbit Ig (Dako, Glostrup, Denmark) at 1:100 dilution, and immunoreactivity was detected by a DAB reaction.

Immunoreactivity of cytoplasmic dynein-1 or α-ubulin was analyzed using Image-Pro plus version 5.0 (Media Cybernetics, Inc., Silver Spring, MD, USA), and the area × density of immunoreactivity in the podocytes in each glomerulus was calculated in 60 total glomeruli cut in the center.

Immunofluorescence were performed in paraffine embedded sections using direct Alexa Fluor labeled antibodies against nephrin (Alexa 680) and podocin (Alexa 546) for slit membrane, or caspase 3 (Alexa 488) and FAS-Ligand (Alex 594) for apoptosis (Life Science Marketplace/Szabo Scandic, Wienna, Austria). Multiple fluorescences were observed using Mantra 2 system (Kiko Tech Co., Ltd., Osaka, Japan).

### 4.3. Electron Microscopy (EM)

Part of the cortex of the kidney was fixed with 2.5% glutaraldehyde (TAAB, Berks, England) in 0.2 M cacodylate buffer solution (Wako, Osaka, Japan), and 2-mm^3^ blocks were cut and post-fixed with osmium tetroxide and embedded in epoxy resin (TAAB), as previously described [13]. The semi-thin sections were cut and stained with Toluidine blue to evaluate the glomerular area and number of podocytes in 5 to 10 blocks per mouse. The ultrathin sections were then cut with an ultramicrotome and counterstained with uranium acetate and lead citrate and observed under a transmission electron microscope (JEM-1011: JEOL, Tokyo, Japan).

### 4.4. Quantitative Real-Time PCR

Total RNA from kidneys was extracted using TRIZOL reagent (Invitrogen, Walthum, MS, USA) as described previously [45]. One microgram of total RNA per sample was subjected to reverse transcription to obtain single-stranded complementary DNA using the TaKaRa RNA PCR Kit Ver. 3.0 (TaKaRa BIO, Shiga, Japan) with a TaKaRa PCR Thermal Cycler 480. Complementary DNA levels of mouse *tubulin α1a*, and *dync1h* were measured using Taq Man Universal Master Mix 2 or PowerUp SYBR Green Master Mix with an Applied Biosystems 7300 Real-Time PCR System (Thermo Fisher Scientific, Tokyo, Japan). The primers used in the present study are shown in Table 1. Relative quantification values of the target were normalized to the value of GAPDH as an endogenous control.

### 4.5. Statistical Analyses

The data were expressed as the mean ± standard error. An analysis of variance was used for statistical comparisons between two groups, followed by a Bonferroni post-hoc analysis. *p* values of <0.05 were considered to indicate statistical significance.

## 5. Limitations

In this study, we were unable to show direct molecular interactions between APC, cytoplasmic dynein 1, α-tubulin and podocyte vesicles. Recently, the intracellular signaling pathway of glomerular filtration barrier, actin, and Rho GTPase has been elucidated [46], and the signaling pathway of transcytosis in podocyte vesicles should be evaluated in future studies. It is also a limitation of this study that we could not harvest enough APC1638T mice for this experiment; thus, the repeated study needs to have enough animals for statistics or to use some other studies blocking APC C terminal by the antibody to confirm our observation.

## 6. Conclusions

It is possible that APC C terminal mutant mice could decrease albuminuria associated with a decrease in podocyte vesicle transport with a reduction in cytoplasmic dynein-1 and α-tubulin when minimal change nephrotic syndrome is induced.

## Figures and Tables

**Figure 1 ijms-22-13412-f001:**
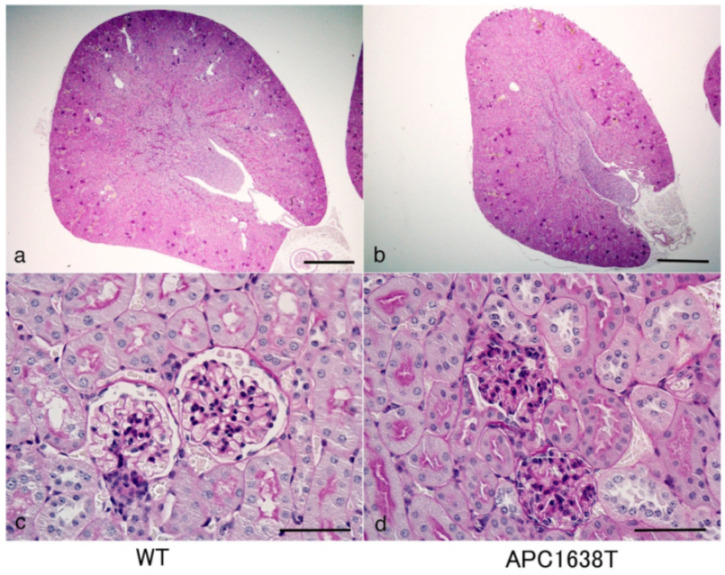
Periodic acid Schiff (PAS) staining of the kidney of wild-type (WT, (**a**,**c**)) and APC1638T mice (**b**,**d**). The bars indicate 500 μm (**a**,**b**) and 50 μm (**c**,**d**).

**Figure 2 ijms-22-13412-f002:**
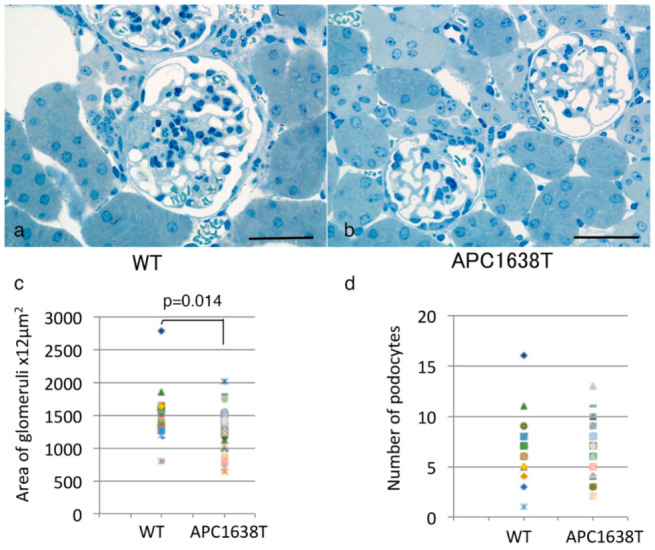
Toluidine blue staining of EPON-embedded renal sections from wild-type (**a**) and APC1638T mouse (**b**). The bars indicate 50 μm (**a**,**b**). The area of glomerular tuft were measured using Image Pro software (**c**), and the number of podocytes (**d**) were measured by counting nucleus of podocyte in each glomerulus from a wild-type mouse (n = 32 glomeruli) and an APC1638T mouse (n = 51 glomeruli).

**Figure 3 ijms-22-13412-f003:**
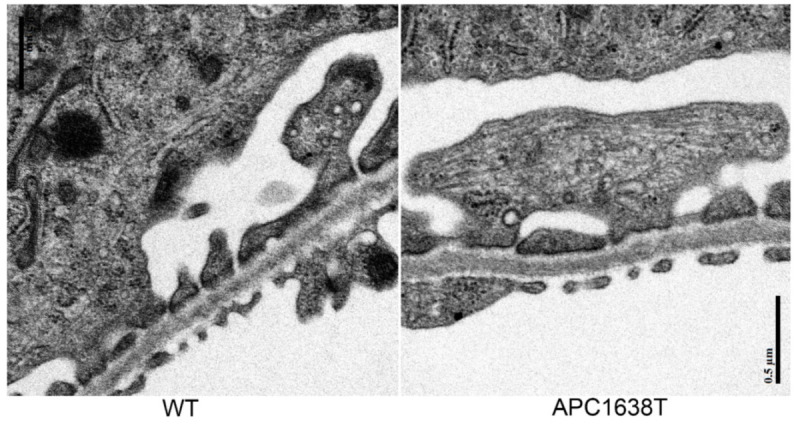
Electron micrograph of podocyte of wild-type (WT) and APC1638T mice. The bar indicates 0.5 μm.

**Figure 4 ijms-22-13412-f004:**
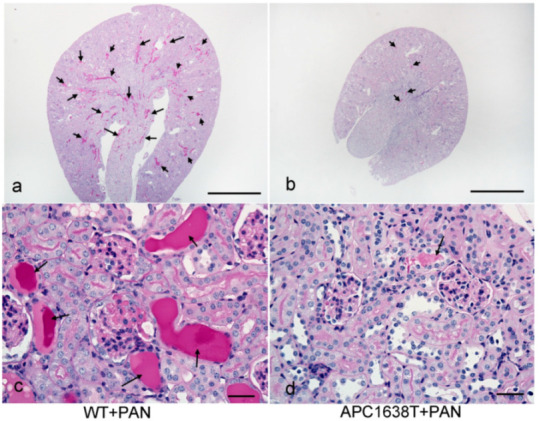
Periodic acid Schiff (PAS) staining of the kidney of wild-type+PAN nephrotic mouse (**a**,**c**) and APC1638T+PAN mouse (**b**,**d**). The bars indicate 500 μm (**a**,**b**) and 50 μm (**c**,**d**). The arrows indicate hyaline casts.

**Figure 5 ijms-22-13412-f005:**
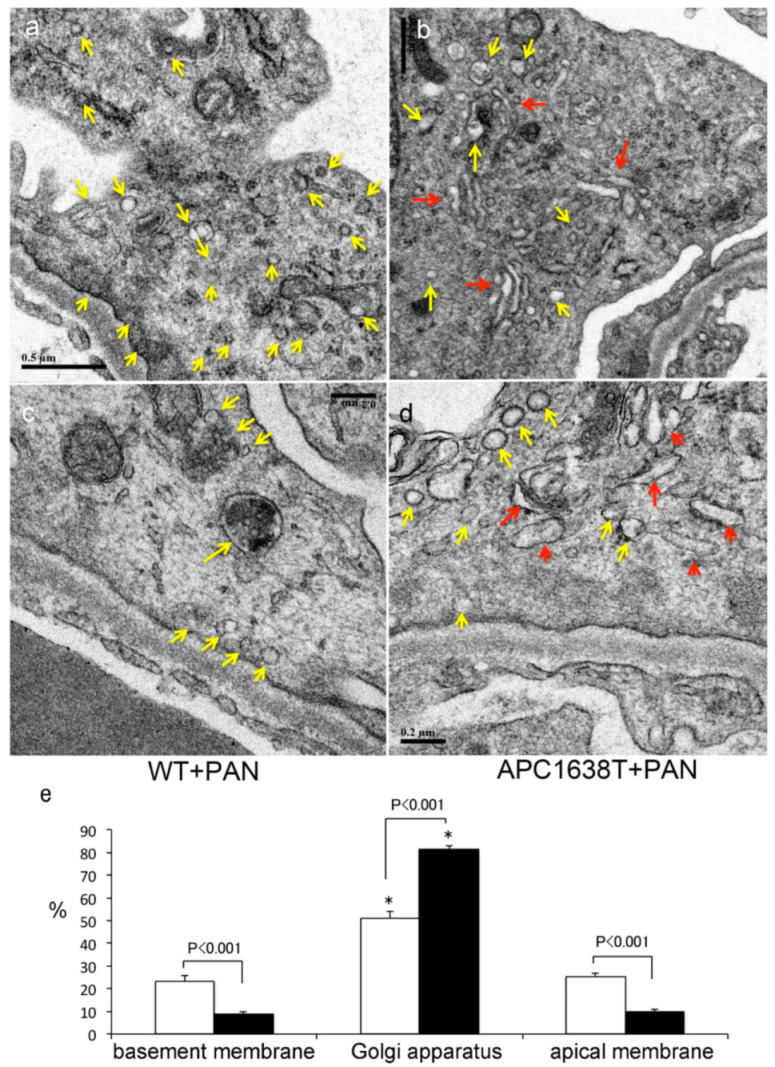
Electron micrograph (**a**–**d**) of podocyte and percent vesicle distribution (**e**) around the basement membrane, Golgi apparatus, and apical membrane of PAN-induced nephrotic syndrome in the wild-type mice (WT+PAN, white bar) and APC1638T mice (APC1638T+PAN, black bar). The yellow arrows indicate endocytosis and transcytosis vesicles, and the red arrows tubular structures related to Golgi apparatus. The bar indicates 0.5 μm (**a**,**b**) and 0.2 μm (**c**,**d**). n = 25 podocytes from 3 WT+PAN mice and n = 32 podocytes from 3 APC1638T+PAN mice. * *p* < 0.001 vs. basement membrane in each mouse.

**Figure 6 ijms-22-13412-f006:**
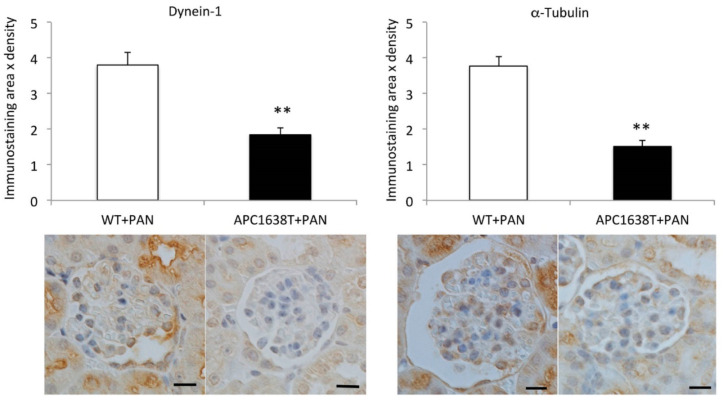
Immunostaining for cytoplasmic dynein-1 and α-tubulin in the kidney of PAN-induced nephrotic syndrome in the wild-type mice (WT+PAN) and APC1638T mice (APC1638T+PAN). Immunoreactivity area and density were analyzed by Image-Pro plus software and summarized as bar graphs. n = 60 glomeruli in each group of 3 mice ** *p* < 0.001 vs. WT+PAN. The bar indicates 10 μm.

**Figure 7 ijms-22-13412-f007:**
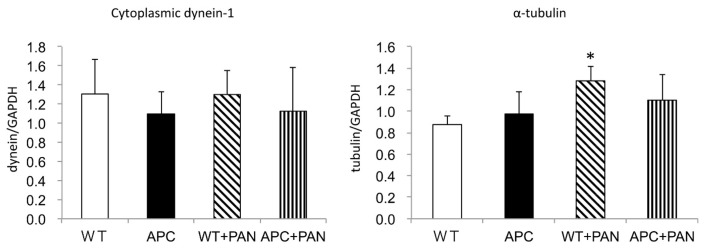
The mRNA expression of cytoplasmic dynein-1 and α-tubulin in the kidney of wild-type mice (WT), APC1638T mice (APC), and their PAN-induced nephrotic syndrome (WT+PAN, APC+PAN). n = 2 measurements from a mouse of WT and APC, n = 4 measurements from 3 mice of WT+PAN and APC+PAN. * *p* < 0.05 vs. WT.

**Figure 8 ijms-22-13412-f008:**
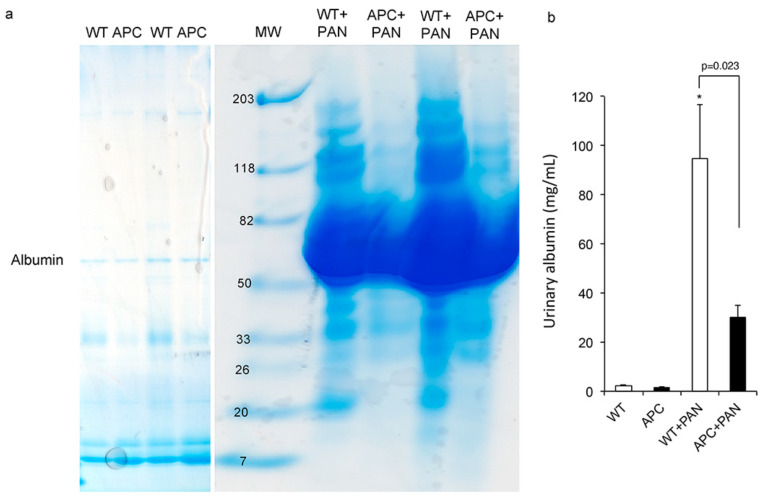
SDS-PAGE analysis of urinary protein (**a**) and densitometry for urinary albumin (**b**) of the wild-type mice (WT), APC1638T mice (APC), and their PAN-induced nephrotic syndrome (WT+PAN, APC+PAN). n = 3 measurements from 2 mice in WT and APC, n = 4 measurements from 3 mice in WT+PAN and APC+PAN. * *p* = 0.0034 vs. WT.

**Figure 9 ijms-22-13412-f009:**
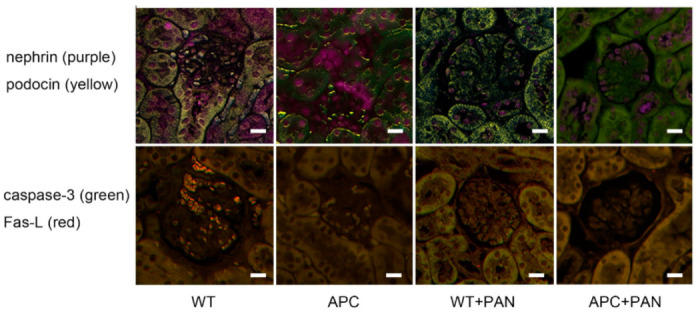
Immunofluorescence for nephrin (purple) and podocin (yellow) in the upper panels, caspase-3 (green) and Fas-Ligand (red) in the lower panels. WT indicates wild-type mice (WT), APC, APC1638T mice, WT+PAN, wild-type mice with PAN-induced nephrotic syndrome, and APC+PAN, APC1638T mice with PAN-induced nephrotic syndrome. The bar indicates 10 μm.

**Figure 10 ijms-22-13412-f010:**
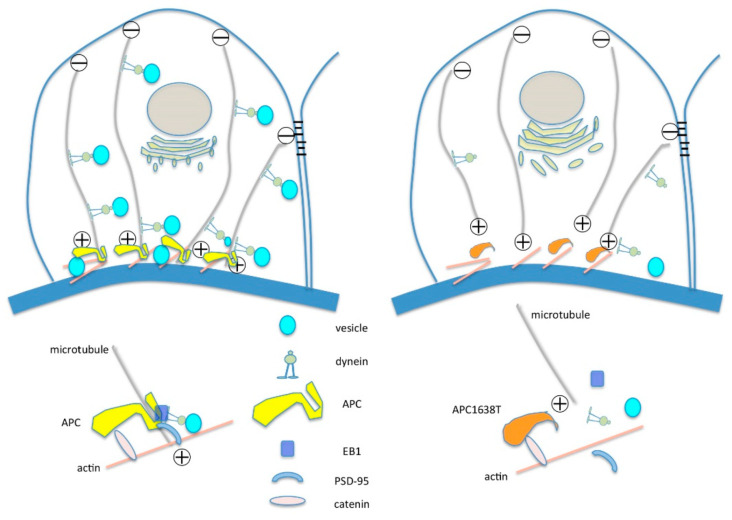
Illustration of molecular bindings of APC C-terminal with microtubule plus-end, PSD-95, EB-1 and cytoplasmic dynein-1 in the WT mice and APC1638T mice.

**Table 1 ijms-22-13412-t001:** Primers for real-time PCR.

Gene	Sequences 5′ to 3′
*Dync1h*	Forward AGTCACAGGTCTGAAGCTCC
	Reverse ACTGTGGAGATGGCATTGGA
*tubulinAlpha1a*	Forward AGCGGCTCTCTGTGGATTAC
	Reverse CAACCACAGCAGTGGAAACC
*GAPDH*	Forward ACCCAGAAGACTGTGGATGG
	Reverse GGATGCAGGGATGATGTTCT

## Data Availability

The data used to support the findings of this study are available from the corresponding author upon request.

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
