# Peer review of "Decreased Podocyte Vesicle Transcytosis and Albuminuria in *APC* C-Terminal Deficiency Mice with Puromycin-Induced Nephrotic Syndrome"

_ijms, 2021, doi:10.3390/ijms222413412_

Round 1
Reviewer 1 Report
The authors studied the minimal change nephrotic syndrome model in mice mutated for the Adenomatous polyposis coli gene. Is it possible to see a representative image of the expression of APC in wild type mice and healthy controls?
In the manuscript, the results are meager. The title states that there is a decrease in podocyte vesicles in the experimental model. However, it is not clear how the authors could make such a claim.
The methodology is equally lacking. How is it possible to count the number of podocytes using a Toluidine blue histological stain? Glomerular cells are of different types, there are not only podocytes.
The number of animals that were used in each experimental group is not mentioned.
The statistics are equally unclear. how many urinary densitometries have been performed? How many images was the area of glomerular tuft evaluated? Some results have the p <0.02, others 0.005, others, 0.05 .... It would be more correct to enter the value of p.
These deficiencies make it difficult to reproduce the results.
there are only many representative images. the most important representative images are missing.
For these reasons the manuscript in my opinion is not suitable for publication in this journal.
Author Response
Answer to the reviewer 1.
The authors studied the minimal change nephrotic syndrome model in mice mutated for the Adenomatous polyposis coli gene. Is it possible to see a representative image of the expression of APC in wild type mice and healthy controls?
Answer: Our coauthor, Prof. Senda has already published the localization of APC in the kidney in Acta Histochem Cytochem 33(6):457-463, 2000. APC is highly expressed through the medullary region of the kidney, and localized mainly in the basal cytoplasm of epithelial cells.
In the manuscript, the results are meager. The title states that there is a decrease in podocyte vesicles in the experimental model. However, it is not clear how the authors could make such a claim.
Answer: Thank you very much for the important comment. As shown in Figure 5 e, podocyte vesicles were significantly decreased from the basal membrane area of endocytosis and apical membrane area of exocytosis, but stayed around Golgi apparatus area of vesicle synthesis. As the reviewer 1 pointed out we changed the title as “Decreased podocyte vesicle transcytosis and albuminuria in APCC-terminal deficiency mice with puromycin-induced nephrotic syndrome”
The methodology is equally lacking. How is it possible to count the number of podocytes using a Toluidine blue histological stain? Glomerular cells are of different types, there are not only podocytes.
The number of animals that were used in each experimental group is not mentioned.
Answer:The specialist of renal morphology (AT) counted the number nuclei of podocyte by toluidine blue staining of EPON-embedded renal sectionsPodocytes were easily identified by the location of outside of capillary wall, endothelial cells located inside of capillary, and mesangial cells were surrounded by mesangial matrix. In this study model, there are no crescents and inflammatory cells infiltration, it is easily distinguish podocytes from endothelial cell or mesangial cells by light microscope. We add the number of animals in the methods.
The statistics are equally unclear. how many urinary densitometries have been performed? How many images was the area of glomerular tuft evaluated? Some results have the p <0.02, others 0.005, others, 0.05. It would be more correct to enter the value of p.
Answer: We add number of samples in the legend of Figure 8 and also 60 glomeruli in each group in Figure 6. According to the reviewer’s comments we expressed p=0.0014 in Figure 2.
Reviewer 2 Report
The authors addressed all the concerns raised from the first submission and significantly improved their manuscript, which may be published in the present form.
Author Response
Answer to the reviewer 2
The authors addressed all the concerns raised from the first submission and significantly improved their manuscript, which may be published in the present form.
Answer: I really appreciate the reviewer 2 for the valuable comments and suggestions.
Reviewer 3 Report
In the current research, the authors investigated the morphological changes in podocyte microtubule and vesicle transport in APC1638T mice with PAN-induced MCNS and discussed the importance of podocyte vesicle
transport as a mechanism of selective albuminuria.
Please correct the formatting of the article, you are not adhering to the IJMS template. It is extremely difficult to follow the text.
Rephrase the title, it is ambiguous.
Correct the affiliation - the country is missing?
line 15: p<0.02 - please correct the way in which you present P-values. Either state P<0.001 or the exact value if >0.001 but <0.05.
It is not clear to me if this paper was previously reviewed & rejected then resubmitted. There are highlighted sections throughout the paper and different font colours. Please clarify. If it was resubmitted, you should have also presented the revision letter addressing the comments of the previous peer-reviewers.
"We recently reestablished the concept of albumin filtration through the podocyte cell body in minimal change nephrotic syndrome (MCNS)" - you are citing a paper from 2008 which was nearly 14 years ago... this is obviously not a recent discovery.
line 53 - why is the font colour different (gray vs black)?
Figure 2: p<0.02 - please correct the way in which you present P-values. Either state P<0.001 or the exact value if >0.001 but <0.05.
line 86: appratus - please correct the spelling of Golgi apparatus
Figure 8: p<0.05, P<0.005 - please correct the way in which you present P-values. Either state P<0.001 or the exact value if >0.001 but <0.05.
lines 167-169: why is the font colour different (gray vs black)?
lines 170-171: why is the font colour different (gray vs black)?
Which are the strengths and limitations of your study? Please discuss them in a separate paragraph before the conclusions.
Which are the conclusions of the study? Please mention them.
The references are extremely old - only 4 references from 2018 onwards. Please update with novel findings regarding the investigated topic.
Author Response
Answer to the reviewer 3
In the current research, the authors investigated the morphological changes in podocyte microtubule and vesicle transport in APC1638T mice with PAN-induced MCNS and discussed the importance of podocyte vesicle transport as a mechanism of selective albuminuria.
Please correct the formatting of the article, you are not adhering to the IJMS template. It is extremely difficult to follow the text.
Answer: We used the IJMS template and rewrite the text again.
Rephrase the title, it is ambiguous.
Answer: We change the title as “Decreased podocyte vesicle transcytosis and albuminuria in APCC-terminal deficiency mice with puromycin-induced nephrotic syndrome”
Correct the affiliation - the country is missing?
Answer: we added the country.
line 15: p<0.02 - please correct the way in which you present P-values. Either state P<0.001 or the exact value if >0.001 but <0.05.
Answer: We change it to p=0.014.
It is not clear to me if this paper was previously reviewed & rejected then resubmitted. There are highlighted sections throughout the paper and different font colours. Please clarify. If it was resubmitted, you should have also presented the revision letter addressing the comments of the previous peer-reviewers.
Answer: I understand that this is a 3rdrevision of my original paper. I attached the answers to the reviewers of 1strevision below with blue color. At the second revision the manuscript number has been changed to a new number, then it become complicated.
Answers to the reviewers (first revision)
Oct.21, 2021
The authors studied the minimal change nephrotic syndrome model in mice mutated for the Adenomatous polyposis coli gene. The authors observed a decrease in albuminuria and the number of podocyte vesicles. Starting from these two results affirmed by the authors in the title, I ask for Major revisions to the manuscript before resubmitting it.
Answer: I really appreciate the reviewer for the careful evaluation of our paper and valuable suggestions. According to the comments, we revised the paper with new data.
- The authors showed representative figures of the images of transcellular vesicles and albuminuria.Is it possible to have quantification and statistical analysis to assert the reduction of albuminuria and podocyte vesicles?
Answer: Thank you very much for the valuable comments. We counted the number of vesicles around basement membrane, Golgi apparatus, apical membrane and the express as percentage of vesicle distribution in Figure 5. We also measure amount of albumin excretion by densitometry in Figure 8.
2) Have proteinuria, lipidemia, cholesterolemia, and albuminemia been investigated? These alterations of the experimental model of minimal lesion nephrotic syndrome should be evaluated.
Answer: We could not get blood samples of these animals and we could not measure serum albumin, cholesterol and lipids, and the amount of urine was too small, thus we used SDS-PAGE to measure urinary albumin and other proteins. We did semi-quantification of urinary albumin by densitometry of the bands. Even though the levels of cholesterol is low in the rodent compared to those in human, it has been already reported that total cholesterol, LDL, lipoprotein were increased and serum albumin decreased with negative correlation to cholesterol in puromycin-induced nephrotic syndrome (Metabolism 38(5):491-495, 1989).
3) Was Nephrin Expression Evaluated? Evaluating the distribution of this molecule, synaptopodin, and podocin would be useful for understanding the distribution of slit diaphragm proteins; of course, it would be a way to quantify the podocytes too.
Answer: It is a nice suggestion and we performed immunofluorescence staining for nephrin and podocin (new Figure 9). There were no difference of nephrin and podocin expression between WT and APC1638T mice in the normal condition and they were reduced after PAN-induced nephrotic syndrome in both WT and APC1638T mice. The reduction of nephrin and podocin after PAN nephrotic syndrome induction reflected the reduction of slit diaphragm by foot process effacement. However, there was no difference in the expression of nephrin or podocin in both WT and APC1638T mice, indicating that the nephrin expression did not related to the reduction of albuminuria in the APC+PAN mice compared to WT-PAN mice.
4) The quantification of podocytes by histological counts is not clear to me. Is it a method that the authors have developed? How do the authors distinguish podocytes from glomerular endothelial cells and mesangial cells in histology?
Answer: The number of podocyte nuclei was counted in the all glomeruli cut near center in each kidney. Podocytes were identified by the location of outside of capillary, endothelial cells located inside of capillary, and mesangial cells were surrounded by mesangial matrix by PAS staining.
5) Line 191 page 16: How was it shown that the observed vesicles had albumin in them?
Answer: We have previously reported that the podocyte vesicles contain albumin molecule by immuno electron microscope. Ref. 7. Tojo, A.; Onozato, M.L.; Kitiyakara, C.; Kinugasa, S.; Fukuda, S.; Sakai, T.; Fujita, T. Glomerular albumin filtration through podocyte cell body in puromycin aminonucleoside nephrotic rat. Med. Mol. Morphol. 2008, 41, 92-98. Ref. 8. Kinugasa, S.; Tojo, A.; Sakai, T.; Tsumura, H.; Takahashi, M.; Hirata, Y.; Fujita, T. Selective albuminuria via podocyte albumin transport in puromycin nephrotic rats is attenuated by an inhibitor of NADPH oxidase. Kidney Int. 2011, 80,1328-1338.
6) Authors should discuss the results shown more. Statistically significant results are the decrease of dynein and alpha-tubulin in PAN-APC1638T mice. For what reason? Why might dynein be a target for albuminuria (line 193, page 11)? Why didn’t you mention these results on abstract?
Answer: Thank you very much. We added the result of dynein-1 and alpha-tubulin in the abstract and discussed that these molecules formed cluster to transport vesicles, whereas these molecular cluster formation may be disturbed in the mice with APC C-terminal mutation.
7) Apart from the reduced size of the glomeruli observed. I advise the authors to describe what kind of activity takes place inside the glomeruli. Are there any apoptotic phenomena or cell proliferation that can be described with immunohistochemical staining with TUNEL and BrdU?
Answer: Thank you very much for the valuable comments. Apoptosis did not occurred because the nuclei of the podocyte did not become smaller by EM observation, and we performed immunostaining for caspase-3 and Fas-L in the new Figure 9, which did not show the significant change between WT and APC1638T mice and their PAN-induced model.
8) I recommend that the authors add a group of healthy mice compared to the other two groups.
Answer: We added the Figure of urinary proteins, nephrin, and podocin in the normal condition of WT and APC1638T mice.
9) I recommend rewriting the discussion more clearly. It is very dispersive.
Answer: We added discussion about dynein and microtubule.
"We recently reestablished the concept of albumin filtration through the podocyte cell body in minimal change nephrotic syndrome (MCNS)" - you are citing a paper from 2008 which was nearly 14 years ago... this is obviously not a recent discovery.
Answer: We omitted the “recently”, and added instruction with some references.
line 53 - why is the font colour different (gray vs black)?
Answer: We changed the color to black.
Figure 2: p<0.02 - please correct the way in which you present P-values. Either state P<0.001 or the exact value if >0.001 but <0.05.
Answer: We revised Figure 2 with p=0.014
line 86: appratus - please correct the spelling of Golgi apparatus
Answer: We fixed the typo.
Figure 8: p<0.05, P<0.005 - please correct the way in which you present P-values. Either state P<0.001 or the exact value if >0.001 but <0.05.
Answer: we revised Figure 8 with p=0.023.
lines 167-169: why is the font colour different (gray vs black)?
Answer: We changed the color to black.
lines 170-171: why is the font colour different (gray vs black)?
Answer: We changed the color to black.
Which are the strengths and limitations of your study? Please discuss them in a separate paragraph before the conclusions.
Answer: We described the limitation of this study.
Limitations
In this study, we were unable to show direct molecular interactions between APC, cytoplasmic dynein 1, α-tubulin and podocyte vesicles.Recently, the intracellular signaling pathway of glomerular filtration barrier, actin, and Rho GTPase has been elucidated [46], and the signaling pathway of transcytosis in podocyte vesicles should be evaluated in future studies.
Which are the conclusions of the study? Please mention them.
Answer: We added the conclusion section.
The references are extremely old - only 4 references from 2018 onwards. Please update with novel findings regarding the investigated topic.
Answer: We added more recent references and rewrote the introduction.
We also refer the recent paper from this journal.
- Asano-Matsuda K.; Ibrahim S.; Takano T.; Matsuda J. Role of Rho GTPase Interacting Proteins in Subcellular Compartments of Podocytes. Int. J. Mol. Sci. 2021, 22, ARTN 3656
10.3390/ijms22073656

Round 2
Reviewer 3 Report
The authors have answered my comments.
Author Response
Answer to the reviewer 3
Thank you very much for evaluation our paper carefully. We have revised the paper according to the comments and also performed native check again.
- Please address the concerns regarding how you determined the identity of the podocytes in the paper itself.
Answer: We have previously published that podocyte has been confirmed by the immunostaining for the podocalyxin (Hanamura K, Tojo A, Urinary and glomerular podocytes in patients with chronic kidney diseases. (Hanamura K, Tojo A, Clin. Exp. Nephrol 18:95-103, 2014)
Please note that "n" is equal to the number of mice studied, not the number of regions of interest that were examined on any particular slide, so n = 52 probably is equal to n = 4 or less.
Answer: According to the reviewer’s comment we have revised the figure legend such asn=25 podocytes from 3 WT+PAN mice and n=32 podocytes from 3 APC1638T+PAN mice.
If only n = 4 mice were studied in each group for each genotype, you may need to soften some conclusions and make mention of the small "n"
Answer: According to the reviewer’s comment, we revised the limitation with small number of animals, and soften conclusion.
“It is also a limitation of this study that we could not harvest enough number of APC1638T mice for this experiment, thus, the repeated study to have enough number of animals for statistics or some other studies blocking APC C terminal by the antibody to confirm our observation.
Conclusions: It is possible that APC C terminal mutant mice could decrease albuminuria associated with a decrease in podocyte vesicle transport with a reduction in cytoplasmic dynein-1 and α-tubulinwhen minimal change nephrotic syndrome was induced.”
Two-way ANOVA statistics might be the method of preference with genotype X treatment and it's interactions being evaluated. Please revise accordingly if possible in this respect.
Answer: we did not applied two-way analysis ANOVA, but used one-way analysis ANOVA in this case.
This manuscript is a resubmission of an earlier submission. The following is a list of the peer review reports and author responses from that submission.
Round 1
Reviewer 1 Report
The authors studied the minimal change nephrotic syndrome model in mice mutated for the Adenomatous polyposis coli gene. The authors observed a decrease in albuminuria and the number of podocyte vesicles. Starting from these two results affirmed by the authors in the title, I ask for Major revisions to the manuscript before resubmitting it:
1) The authors showed representative figures of the images of transcellular vesicles and albuminuria. Is it possible to have quantification and statistical analysis to assert the reduction of albuminuria and podocyte vesicles?
2) Have proteinuria, lipidemia, cholesterolemia, and albuminemia been investigated? These alterations of the experimental model of minimal lesion nephrotic syndrome (10.1038/srep32087) should be evaluated.
3) Was Nephrin Expression Evaluated? Evaluating the distribution of this molecule, synaptopodin, and podocin would be useful for understanding the distribution of slit diaphragm proteins; of course, it would be a way to quantify the podocytes too.
4) The quantification of podocytes by histological counts is not clear to me. Is it a method that the authors have developed? How do the authors distinguish podocytes from glomerular endothelial cells and mesangial cells in histology?
5) Line 191 page 16: How was it shown that the observed vesicles had albumin in them?
6) Authors should discuss the results shown more. Statistically significant results are the decrease of dynein and alpha-tubulin in PAN-APC1638T mice. For what reason? Why might dynein be a target for albuminuria (line 193, page 11)? Why didnt you mention these results on abstract?
7) Apart from the reduced size of the glomeruli observed. I advise the authors to describe what kind of activity takes place inside the glomeruli. Are there any apoptotic phenomena or cell proliferation that can be described with immunohistochemical staining with TUNEL and BrdU?
8) I recommend that the authors add a group of healthy mice compared to the other two groups.
9) I recommend rewriting the discussion more clearly. It is very dispersive.
Reviewer 2 Report
The present study by Dr. Hatakeyama and co-authors investigates the role of adenomatous polyposis coli C-terminal mutation (APC1693T) in PAN-induced nephrotic syndrome. The data provide convincing evidence that APC1693T mice with PAN-induced nephrotic syndrome have fewer of endocytosis and transcytosis vesicles compared to wildtype mice with PAN-induced nephrotic syndrome, while APC1693T mice are protected from albuminuria. The work is based on in vivo data only. Unfortunately, this work does not provide a mechanistical support of an observed facts giving an explanation why APC1693T mice are protected from PAN-induced albuminuria, and misses some very important controls. All the data presented for the experiment with PAN-induced nephrotic syndrome compare only WT+PAN and APC1693T+PAN groups. Is there a difference between WT and WT+PAN groups or APC1693T and APC1693T+PAN groups? Is there a difference in foot processes effacement and/or podocyte number between APC1693T and APC1693T+PAN? Do podocytes from APC1693T+PAN mice have less vesicle transport or differences in Dynein-1/alfa-tubulin expression compared to APC1693T mice? It is also not clear if APC1693T mice are globally mutated or it is a podocyte-specific mutation. Since the study aims to investigate podocyte vesicle transport, it is questionable why the authors report data using whole-kidney lysate (Figures 6 and 7) and not from isolated glomeruli (ideally, from primary cultured podocytes). It would be more informative to perform urine albumin/creatinine ELISA to get a quantitative answer if APC1693T+PAN mice have less albuminuria compare to wildtype mice. Also, serum BUN and creatinine levels would give an additional information on kidney function in APC1693T+PAN mice compared to APC1693T and wildtype mice. Some minor concerns have also raised:
- In the Abstracts, please correct that APC1693T mice with PAN-induced nephrotic syndrome have “suppressed urinary albumin excretion” (line 21-22).
- Use better TEM picture for APC1693T mice (Figure 3), which also includes glomerular basement membrane and fenestrated epithelium.
- Add arrows showing hyaline casts on Figure 4.
- Add Y-axis labels on Figure 7. Please explain what “*p<0.05, **p<0.001 from 0” means.
- Add information about age/sex of mice used in the experiment.
- Grammar/spelling checking is required.